# Effect of Gas Diffusion Layer Thickness on the Performance of Anion Exchange Membrane Fuel Cells

Van Men Truong [1],*, Ngoc Bich Duong [1] and Hsiharng Yang [2,3],*

1   School of Engineering and Technology, Tra Vinh University, Tra Vinh 87000, Vietnam;
    ngocbich1184@tvu.edu.vn
2   Graduate Institute of Precision Engineering, National Chung Hsing University, 145 Xingda Road,
    South District, Taichung City 402, Taiwan
3   Innovation and Development Center of Sustainable Agriculture (IDCSA), National Chung Hsing University,
    Taichung City 402, Taiwan
*   Correspondence: tvmen@tvu.edu.vn (V.M.T.); hsiharng@dragon.nchu.edu.tw (H.Y.)

**Abstract:** Gas diffusion layers (GDLs) play a critical role in anion exchange membrane fuel cell (AEMFC) water management. In this work, the effect of GDL thickness on the cell performance of the AEMFC was experimentally investigated. Three GDLs with different thicknesses of 120, 260, and 310 µm (denoted as GDL-120, GDL-260, and GDL-310, respectively) were prepared and tested in a single $H_2/O_2$ AEMFC. The experimental results showed that the GDL-260 employed in both anode and cathode electrodes exhibited the best cell performance. There was a small difference in cell performance for GDL-260 and GDL-310, while water flooding was observed in the case of using GDL-120 operated at current densities greater than 1100 mA cm$^{-2}$. In addition, it was found that the GDL thickness had more sensitivity to the AEMFC performance as used in the anode electrode rather than in the cathode electrode, indicating that water removal at the anode was more challenging than water supply at the cathode. The strategy of water management in the anode should be different from that in the cathode. These findings can provide a further understanding of the role of GDLs in the water management of AEMFCs.

**Keywords:** anion exchange membrane fuel cell; gas diffusion layer; GDL thickness; fuel cell performance

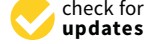

## 1. Introduction

Anion exchange membrane fuel cells (AEMFCs) have been an interesting topic as this fuel cell technology is expected to be able to replace proton exchange membrane fuel cells (PEMFCs) owing to their possibility of using non-precious metals as catalysts and low-cost electrolytes [1–3], reducing the cost issue of fuel cell devices. Additionally, AEMFCs also provide advantages including lower overpotential electrode reactions [4,5], lower fuel crossover [6], a less-corrosive environment [7], and more choices of fuels, including hydrogen gas or nitrogen-based fuels such as ammonia liquid and urea [8–10]. Typical schematic diagrams of an AEMFC and a PEMFC composed of anode and cathode electrodes with an anion exchange membrane or proton exchange membrane for conducting OH$^-$ or H$^+$ ions, respectively, in between are shown in Figure 1 [11]. Although the designs of AEMFC and PEMFC units are similar, their working principles are different due to the different electrolytes in AEMFCs and PEMFCs, which leads to their distinct oxidation/reduction (redox) reactions. In particular, the redox reactions of PEMFCs as H$^+$ ions passing through the acidic membrane are shown as follows. The hydrogen oxidation reaction (HOR) and the oxygen reduction reaction (ORR) take place at the anode and the cathode, respectively.

At anode: $2H_2 \rightarrow 4H^+ + 4e^-$;

At cathode: $O_2 + 2H^+ + 4e^- \rightarrow 2H_2O$;

Overall: $2H_2 + O_2 \rightarrow 2H_2O$.

On the other hand, the redox reactions of AEMFCs as $OH^-$ ions passing through the basic membrane are presented as follows:

At anode: $2H_2 + 4OH^- \rightarrow 4H_2O + 4e^-$;

At cathode: $O_2 + 2H_2O + 4e^- \rightarrow 4OH^-$;

Overall: $2H_2 + O_2 \rightarrow 2H_2O$.

Obviously, as compared with PEMFCs in which water is produced at the cathode, the water management in AEMFCs is considerably more complicated because the water production and consumption are simultaneously involved at the anode and cathode, respectively, during operation. Additionally, the issue of water management with the flooding at the anode and dry-out at the cathode can be encountered due to the unbalanced condition of electro-osmosis and water diffusion within the membrane. Therefore, it is necessary to ensure that there is both an adequate quantity of water in the membrane and that flooding/dry-out issues in the anode/cathode electrodes are avoided during the AEMFC operation [12].

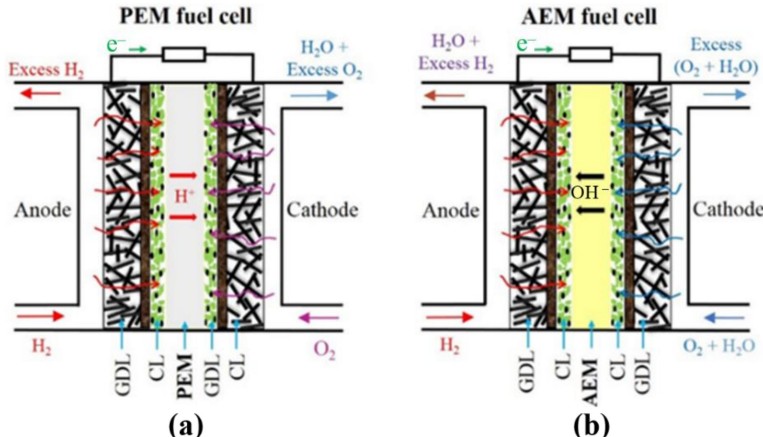

**Figure 1.** Typical Schematic diagrams of (**a**) PEMFC and (**b**) AEMFC.

In an AEMFC unit, gas diffusion layers (GDLs) play a key role in reactant gas diffusion and water management. In the recently developed fuel cells, a GDL is usually a sheet structure formed by coating a microporous layer (MPL) on a gas diffusion substrate (GDS). An MPL is composed of carbon microparticles and hydrophobic fluoropolymer, while the common GDS materials include nonwoven carbon paper, non-woven carbon felt, or woven carbon cloth because of their good electrical conductivity, high porosity, stability, and durability. Furthermore, in order to achieve a better fuel cell performance by optimizing the water transport, the carbon fiber in the GDS is coated with hydrophobic agents such as polytetrafluoroethylene (PTFE) [13–16] or fluorinated ethylene propylene (FEP) [16].

Unlike PEMFCs with numerous studies on GDLs [17], few attempts have been carried out in the field of AEMFCs. For example, in the study of Y.S. Li et al. [18], the influence of the MPL on the cell performance of an anion-exchange membrane direct ethanol fuel cell was experimentally investigated. The cell performance was considerably upgraded as an MPL was inserted between the catalyst layer (CL) and the GDS in the cathode electrode. This was because that the water crossover was alleviated by the MPL barrier, and thus the water flooding was decreased. Additionally, the role of the MPL in the water transport during AEMFC operation has also been numerically studied by some research groups [19–21]. The effective diffusivity could be one of the important parameters of MPL properties. Besides, the effects of PTFE content and Pt catalyst loading in the CL on the AEMFC performance was experimentally studied by D. Yang et al. [22]. They reported that the highest cell performance was obtained at the PTFE content of 20% with a catalyst loading of 1.0 mg cm$^{-2}$. Kaspar et al. [23] studied the water conditions in hydroxide exchange membrane fuel cells or AEMFCs by varying some design and operation parameters including GDL wetproofing, MPL insertion, ionomer loading, and reactant gas humidi-

fication. The gas humidification and wetproofing requirements could differ between the anode and the cathode electrodes in order to achieve high cell performance. Additionally, the water permeation through the anion exchange membrane (AEM) is one of the factors influencing water management in AEMFCs. Taking this into account, Luo et al. [24] conducted experiments to study the water permeation through membranes of different thicknesses. Their observations revealed that the degree of liquid–liquid water permeation was much smaller than that of liquid–vapor water permeation through AEMs. In addition, interfacial water permeation through membranes is determined by their interface. More recently, Omasta et al. [25] experimentally investigated the effects of gas diffusion, gas flow rates, and dew points at the anode and cathode sides on the AEMFC performance. They found that the cell performance could be significantly improved by maintaining the water balance conditions in the AEMFC units during their operation. Similarly, the influence of gas humidification on the AEMFC performance was studied by Reshetenk et al. [26]. The highest cell performance can be achieved at 50/50% relative humidity on the anode and cathode electrodes. They concluded that the GDL's texture and composition along with electrode preparation affect the water management in AEMFCs.

Although water management in AEMFCs is predicted to be more complicated than that in PEMFCs, there have been few studies confirming this and, therefore more research is needed. It is known that using properly designed GDLs is key to achieving appropriate water management in AEMFC devices. Among physical properties, the thickness of GDLs also has a considerable influence on cell performance. However, to the best of our knowledge, an experimental study has not yet been conducted to explore the correlation between the GDL thickness and AEMFC performance. Accordingly, this study is motivated by the need to address this gap. Three GDLs with different thicknesses were prepared and tested. Some of the physical properties of the GDLs, which varied with the change of the GDL thickness, were measured to correlate the relationship between the GDL thickness and its respective device performance. The current density-voltage characteristics were recorded using a single AEMFC with the reactant gases $H_2$ and $O_2$ at the anode and the cathode, respectively. By examining how the variations of the cell performance are affected by embedding different GDL thicknesses in the cathode and anode electrodes, it is hoped that a greater understanding of the effect of GDLs on the water management in AEMFCs can be achieved.

## 2. Experimental

### 2.1. The Preparation of Gas Diffusion Layers

Three types of GDL (CeTech Co., Ltd., Tai Chung, Taiwan) composed of the gas diffusion substrate (GDS) and MPL were prepared for AEMFC performance testing. Basically, the GDS was fabricated based on wet paper-making technology. Polyacrylonitrile (PAN) carbon fiber manufactured by the Japanese Toho Corporation in lengths of 3 and 6 mm was selected as the raw material. Polyacrylamide (PAM) dispersant at the concentration of 0.05 wt.% was mixed with the carbon fiber at a proportion of 0.1%. PVA fibers were added to at 10% of the carbon fiber weight to increase the paper strength. The phenolic resin impregnation, carbonization, graphitization, and hydrophobic treatment were post-processing steps to form carbon fiber paper or GDS. The MPL composed of carbon powder, 30 wt.% PTFE, and additives was spin-coated on the prepared gas diffusion substrate (GDS). The MPL thickness was designed to be about 35 μm. The measured MPL thicknesses of GDL-120, GDL-260, and GDL-310 from cross-sectional SEM images were 36.1, 32.1, and 33.8 μm (as seen in Figure 2c,f,i). The differences between the designed MPL thicknesses and the measured thicknesses could be a result of the manufacturing tolerance of MPL thickness (about ±2 μm) and the unsmooth GDS surface (as seen in Figure 2a,d,g). Their physical properties are presented in Table 1. The thicknesses of these samples, namely, GDL-120, GDL-260, and GDL-310 were 120, 260, and 310 μm, respectively. The nominal basic weight and the through-plane resistance increased with the increasing thickness. However, the through-plane air permeability (measured using the Gurley method, ASTM

D737) was not much different for these prepared samples. This was because the thickness and composition of the MPL synthesized were similar for all samples, resulting in similar air permeability resistance. The hydrophobicity of the gas diffusion substrate was obtained by treating with 30 wt.% PTFE, which was based on the results of our previous study [11]. Since the hydrophobic treatment process of the prepared samples was the same, the contact angle which represents the hydrophobicity was similar among samples. The surface morphology of the GDL samples was also observed using scanning electron microscopy (SEM), as shown in Figure 2. From the SEM images, the back surfaces (substrate surfaces) and MPL surfaces were similar among the prepared samples. In addition, these images show that the substrates were mainly made of carbon fibers with PTFE coating on their surfaces while the MPL consisted of carbon particles with PTFE particles. In the cross-sectional images, the microstructure inside the GDLs and their thickness can also be observed.

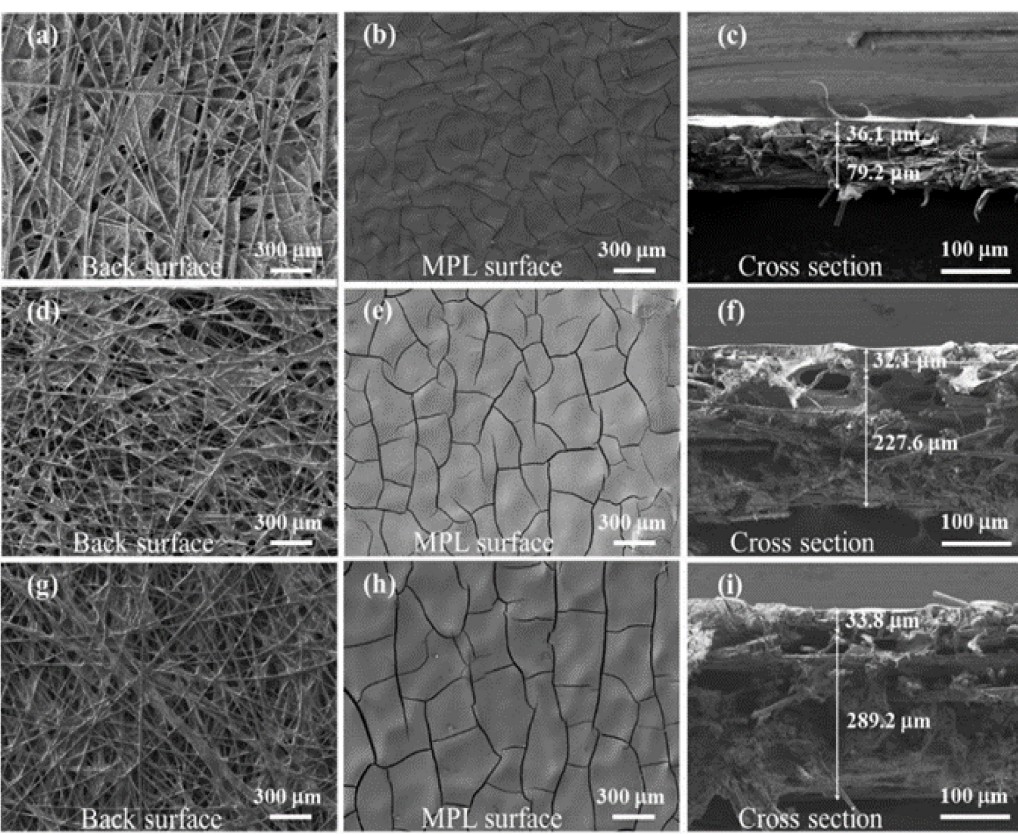

**Figure 2.** Plane-view and cross-sectional SEM images of GDL-120 (**a–c**), GDL-260 (**d–f**), and GDL-310 (**g–i**).

**Table 1.** The properties of the prepared GDL samples.

| Sample | GDL-120 | | GDL-260 | | GDL-310 | |
|---|---|---|---|---|---|---|
| Thickness (μm) | 120 | | 260 | | 310 | |
| Nominal basic weight (g/m$^2$) | 95.3 | | 140.2 | | 150.7 | |
| Air permeability (s) | 98.5 | | 98.9 | | 99.5 | |
| Through-plane resistance (mΩ cm$^2$) | 4.66 | | 7.40 | | 11.19 | |
| Contact angle (°) | MPL | Back | MPL | Back | MPL | Back |
| | 145.4 | 145.6 | 143.1 | 144.7 | 146.2 | 147.5 |

### 2.2. Single AEMFC Assembly and Evaluation

For AEMFC performance evaluation, the membrane electrode assembly (MEA) was made by fabricating the gas diffusion electrodes (GDEs) and then assembled with a commercial anion exchange membrane (AEM). The preparation process was as follows: Firstly, the catalyst ink was prepared by dispersing 40 wt.% Pt/C powder (Tanaka) to a mixture of DI (deionized) water and IPA (isopropyl alcohol) with a volume ratio of 1:1. Next, 20 wt.% aQAPS-$S_{14}$ ionomer (2 wt. % DMF, Alfa Aesar) was added and sonicated for 60 min to get a homogeneous solution. After that, the catalyst ink was coated on the MPL sides of GDL-120, GDL-260, or GDL-310 by hand-brushing on a hot plate at 80 °C to form GDEs. The catalyst loading was 0.8 mg cm$^{-2}$ on both the anode and cathode electrodes. The commercial AEM, namely, AT-1 (purchased from Hephas Energy Co., Ltd., Hsinchu, Taiwan), with a thickness of 40 µm in the dry form was employed. The membrane and the prepared GDEs were immersed in 1 M KOH solution for 48 h to convert the membrane and aQAPS-$S_{14}$ ionomer from chloride form ($Cl^-$) to the hydroxide form ($OH^-$) before making the membrane electrode assemblies (MEAs).

Before testing, the MEAs were made by placing the treated membrane between two above-prepared GDEs without hot pressing. Then, the MEAs were manually installed in a single AEMFC unit consisting of two graphite plates with triple serpentine flow channels (1.0 mm × 1.0 mm × 32 mm) and ribs (1.5 mm × 32 mm), two gold-coated copper current collector plates, and two aluminum endplates. The active electrode area was 10.24 cm$^2$. Two Teflon gaskets with appropriate thickness were also used to provide a 20–30% GDL compression. In particular, with the CL thickness of about 20 µm, the gasket thicknesses of 110 µm, 220 µm, and 250 µm were accompanied with the GDLµ120, GDL-260, and GDL-310, respectively, as the single AEMFC stack was assembled. All these components of the single cell were fixed by 8 screws with 1.4 Nm of torque each screw (Figure 3). Then, the testing cell was connected to a fuel cell testing system (FCED-PD50 test station, Asia Pacific Fuel Cell Technologies, Ltd., Miaoli, Taiwan). During testing, the pure $H_2$ and $O_2$ gases were fed continuously to the anode and cathode at flow rates of 1.0 and 0.5 standard liters per minute (slm), respectively. The cell temperature and $H_2/O_2$ gas dew points were controlled at 70 °C and 65/70 °C, respectively. In addition, for each test, the procedure to achieve the cell stabilization before recording polarization curves was first performed, similar to [27]. In particular, the stabilization was obtained by firstly ramping from 10 to 100 mA cm$^{-2}$ at 10 mA cm$^{-2}$ increments and 1 min per step, followed by a 30 min hold at 100 mA cm$^{-2}$; then slowly ramping from 100 to 400 mA cm$^{-2}$ at 10 mA cm$^{-2}$ increments and 3 min per step, followed by a 30 min hold at 400 mA cm$^{-2}$; and finally, slowly ramping from 400 to 700 mA cm$^{-2}$ at 10 mA cm$^{-2}$ increments and 3 min per step, followed by a 30 min hold at 700 mA cm$^{-2}$. The polarization curves were taken from 1.1 to 0.2 V in 0.02 V increments for each cycle. The analyzed polarization curves were collected from over 50 cycles for each time test at stable operating conditions.

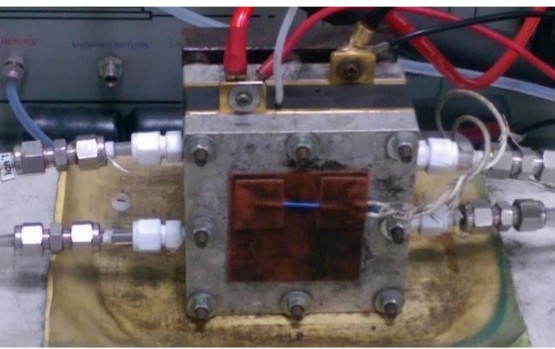

**Figure 3.** A single cell unit prepared for testing.

## 3. Results and Discussion

Figure 4 shows the polarization and power density curves of a single AEMFC for different GDL thicknesses employed in both the anode and cathode electrodes. This single cell was operated at 70 °C under pure $H_2$ and $O_2$ gas supplement at controlled dew points 65 °C and 70 °C, respectively. The cell temperature and gas dew points were set based on our previous study [28] where the operating parameters were optimized. The results show that the GDL-260 exhibited the best cell performance as compared with GDL-120 and GDL-310, as seen in Figure 4. In particular, the peak power densities of the cells using GDL-120, GDL-260, and GDL-310 were 621.7 mW cm$^{-2}$ at 0.41 V, 567.2 mW cm$^{-2}$ at 0.41 V, and 553.5 mW cm$^{-2}$ at 0.49 V, respectively. The lower cell performance of GDL-310 compared to that of GDL-260 can be explained by considering the factors of a thicker GDL impacting upon the cell performance during its operation. It is known that although a thicker GDL provides better mechanical strength to support the catalyst layer and membrane, it has longer diffusion paths for gas and water transport inside the GDL [29] and higher through-plane resistance, both of which cause a negative effect on the fuel cell performance [30]. As presented in Table 1, the through-plane resistance of GDL-310 (11.19 mΩ cm$^2$) was higher than that of GDL-260 (7.40 mΩ cm$^2$), leading to a higher ohmic loss in the cell with GDL-310. In addition, as seen in Figure 3, at the high current density regions, the larger reduction in potential, and then in the power density observed for the GDL-310 as compared with the GDL-260, can be ascribed to the mass transport loss due to greater water accumulation inside the GDL-310, leading to lower mass flow rates of reactant gases passing through the GDL-310. The maximum generated current density for the GDL-310 (1884.2 mA cm$^{-2}$ at ~0.2 V) was also smaller than that for the GDL-260 (2191.1 mA cm$^{-2}$ at ~0.2 V). In contrast, although a thinner GDL has a lower through-plane resistance and a shorter path for gas and water transport, which is beneficial for cell performance, thinner GDLs encounter more flooding than thicker GDLs. This can be interpreted through Lin's observation [25]. Based on the study of the pressure drops across GDLs and water saturation levels inside GDLs, it was found that a thinner GDL will have less space available for reactant gas and water transport due to the smaller pore volume of a thinner GDL compared to a thicker GDL. Therefore, once the same rate of liquid water is pumped into GDLs, the liquid water saturation level of a thinner GDL will be higher than that of a thicker GDL. As a result, flooding may occur more easily in a thin GDL. This was observed in the cell operated with GDL-120. Severe flooding occurred at the current densities greater than 1100 mA cm$^{-2}$. This could be because the rate of water production was faster at higher operating currents, resulting in more water accumulation which occupied most of the pore volume inside the GDL, thereby leading to a significant gas transport resistance. Moreover, from the polarization curve, it was also found that the ohmic loss of the cell with GDL-120 at a current region from 400–1100 mA cm$^{-2}$ was larger than that with GDL-260. As presented in Table 1, the through-plane resistance of GDL-120 (4.66 mΩ cm$^2$) was smaller than that of GDL-260 (7.40 mΩ cm$^2$) but the ohmic loss of GDL-120 was higher than that of GDL-260. This could be due to the lower mechanical strength of GDL-120 compared to GDL-260, causing a higher contact resistance between the catalyst layers and membrane surfaces. Obviously, the GDL thickness is also one of the parameters that needs to be optimized in order to have a balance between positive and negative factors, such as mechanical strength, through-plane resistance, pore volume, water and gas transport path length, etc., which influence the fuel cell performance.

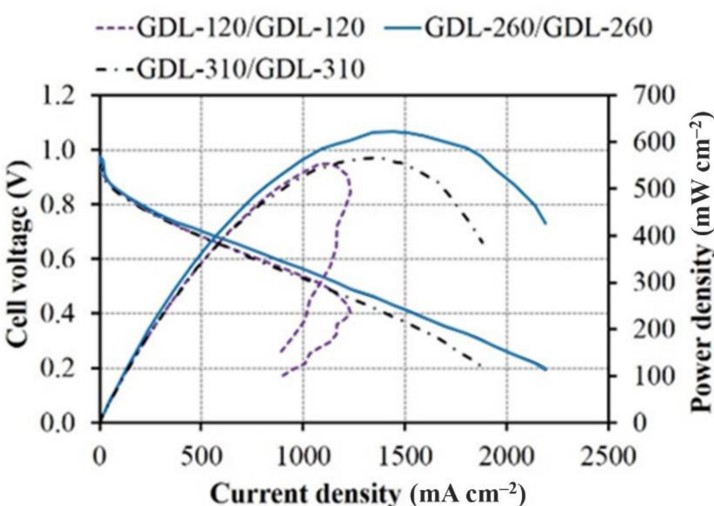

**Figure 4.** Polarization and power density curves for different GDL thicknesses installed in both anode and cathode sites.

To further investigate the influence of the different GDL thicknesses at the anode and cathode electrodes on the cell performance, various MEAs were prepared by employing asymmetrical GDL thicknesses for the anode and cathode electrodes. The polarization and power density curves of these MEAs are presented in Figure 5. The results show that the MEAs consisting of the GDL-260 and GDL-310 in the anode and cathode electrodes, respectively (i.e., GDL-260/GDL-310) or vice versa (i.e., GDL-310/GDL-260) produced lower power densities as compared to the MEA containing GDL-260 on both sides (i.e., GDL-260/GDL-260), indicating that the thickness of GDL-260 was more appropriate for both the anode and cathode electrodes than that of the GDL-310 (Figure 5a). Moreover, the cell performance of the MEA with GDL-260/GDL-310 was better than that with GDL-310/GDL-260, revealing that the thicker GDL-310 located at the anode had a more negative impact on mass transport than when it was located at the cathode. This could be explained through the working principle of AEMFCs. During the operation, water is simultaneously generated and consumed at the anode and cathode sites, respectively. The flow directions of water and reactant gas transport in the anode are opposite, while they are the same in the cathode. Accordingly, the water transport inside the anode GDL could be more difficult than that inside the cathode GDL due to the effect of the kinetic pressure of the gas flow on water transport. In particular, the $O_2$ gas flow could partly assist the water transport on its way to the cathode whereas the $H_2$ gas flow may hinder the water removal at the anode. Consequently, the cell performance will be hampered by a thicker GDL as employed at the anode rather than at the cathode. In addition, severe flooding was still observed for the MEA with GDL-120/GDL-260 while it was slightly reduced for the MEA with GDL-260/GDL-120 (as shown in Figure 5b), indicating that the thinner GDL-120 had a greater impact on mass transport in the anode electrode than in the cathode electrode. This explanation could be similar to the case of thicker GDL used (GDL-310/GDL-260 and GDL-260/GDL-310). These observations confirmed that the water removal at the anode is more challenging than water supplement at the cathode. Therefore, different strategies of water management in the anode and cathode electrodes should be considered.

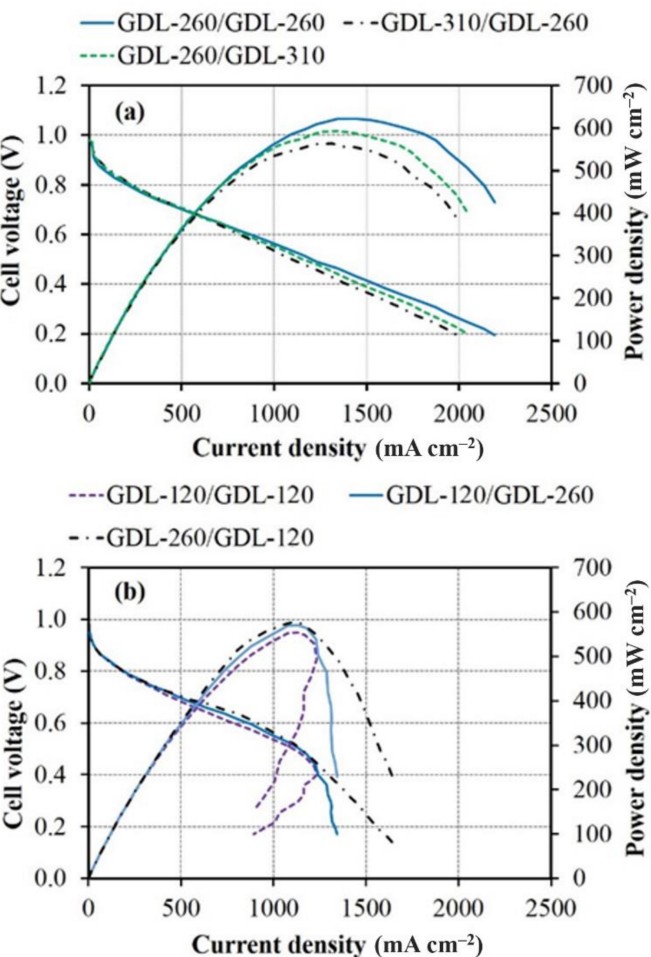

**Figure 5.** The polarization and power density curves of single cell testing using: (**a**) GDL-260/GDL-260, GDL-310/GDL-260, and GDL-260/GDL-310; (**b**) GDL-120/GDL-120, GDL-120/GDL-260, and GDL-260/GDL-120.

## 4. Conclusions

In this work, the effect of GDL thickness on the AEMFC performance was experimentally studied. The results revealed that the GDL thickness had a noticeable influence on mass transport inside the electrodes, and thus on the cell performance. Under the test conditions, the GDL-260 showed a better AEMFC performance than GDL-310 and GDL-120. The lower cell performance of the thicker GDL-310 can be ascribed to the ohmic and mass transport losses due to higher through-plane resistance and longer diffusion paths for water and reactant gas transport. On the other hand, water flooding more easily occurred when using the thinner GDL-120. This could be related to the smaller pore volume of the thinner GDL that is available for water transport. Using inappropriate GDL thickness in the anode tended to hamper the cell performance more severely than in the cathode, indicating that water management in the anode is more challenging than in the cathode. Our findings can provide useful insights for the development of GDLs used in AEMFCs.

**Author Contributions:** Conceptualization, V.M.T. and H.Y.; Data curation, N.B.D.; Formal analysis, N.B.D. and H.Y.; Funding acquisition, H.Y.; Investigation, V.M.T.; Methodology, V.M.T.; Project administration, H.Y.; Supervision, H.Y.; Validation, V.M.T.; Writing—review & editing, V.M.T. and H.Y. All authors have read and agreed to the published version of the manuscript.

**Funding:** This research was funded by the Ministry of Science and Technology, Taiwan under the grants MOST-108-2221-E-005-027 and MOST-108-3116-F-005-002.

**Acknowledgments:** This work was funded by the Ministry of Science and Technology, Taiwan under the grants MOST-108-2221-E-005-027 and MOST-108-3116-F-005-002, and was supported in part by Tra Vinh University.

**Conflicts of Interest:** The authors declare no conflict of interest.

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
