# Peer review of "Effect of Gas Diffusion Layer Thickness on the Performance of Anion Exchange Membrane Fuel Cells"

_processes, doi:10.3390/pr9040718_

Round 1

Reviewer 1 Report

1. line 76
CL should be replaced by:  catalyst layer (CL).

2. line 243
On the other hand, water flooding easily occurs for the case of using the thinner GDL-120. This could be related to a smaller pore volume of the thinner GDL which is available for water transport. 

Comment:
Pore volumes normalized against the weight or the volume of GDLs are probably the same for all studied samples, therefore it is only a matter of time, not properties, when flooding occurs. I would not say it is a vital scientific conclusion.
Pore flooding depends on: hydrophobicity, a mean pores dimenison, and possibly temperature gradient across the GDLs. Unfortunately, the authors do not address these issues in their experiments and discussion.

Author Response

Dear Reviewer,

We would like to thank the reviewer for their thoughtful review of the manuscript and provide valuable comments and suggestions. You raise important issues and your inputs are very helpful for improving the manuscript. We agree with your comments and we have revised our manuscript accordingly. All the changes are marked in red using the “Track Changes” function in Microsoft Word.

We hope that the reviewer will find our responses to their comments satisfactory, and we are willing to finish the revised version of the manuscript including further suggestions that the reviewer may have.

We thank the reviewer’s helpful comments and suggestions and provide the answers to specific comments below.

The attached file is our answers to your comments.

Comment 1:

 line 76: CL should be replaced by:  catalyst layer (CL).

Answer to comment 1:

We thank the reviewer’s comment. This is our mistake. We have edited and the changes are marked in red in the manuscript, line 76.

Comment 2:

line 243: On the other hand, water flooding easily occurs for the case of using the thinner GDL-120. This could be related to a smaller pore volume of the thinner GDL which is available for water transport. 

Pore volumes normalized against the weight or the volume of GDLs are probably the same for all studied samples, therefore it is only a matter of time, not properties, when flooding occurs. I would not say it is a vital scientific conclusion.

Pore flooding depends on: hydrophobicity, a mean pores dimension, and possibly temperature gradient across the GDLs. Unfortunately, the authors do not address these issues in their experiments and discussion.

Answer to comment 2:

We thank the reviewer’s comment. We totally agree with the reviewer that the pore volumes normalized against the weight or the volume of GDLs are probably the same for all studied samples because they were made by the same materials and process. Therefore, when if the same surface area is used to prepare gas diffusion electrodes, the thinner GDL should have a smaller pore volume due to a smaller volume of GDL. In Lin and Nguyen’s study [1], based on the measuring of the pressure drop across GDLs and water saturation level inside GDLs, they concluded that a thinner GDL will have less space available for reactant gas and water transport due to smaller pore volume of a thinner GDL compared to a thicker GDL. As a result, too thin GDL will hamper the performance of the proton exchange membrane fuel cells. In fact, our explanation is basically based on the Lin and Nguyen’s findings (as emphasized in the section of results and discussion) due to a similar situation observed in our study and theirs, although we have not measured the pressure drop across the GDLs. Thus, we believe that our conclusion is not a vital scientific one and we hope the reviewer agrees with it.

We also agree with the reviewer that the flooding phenomenon in fuel cells also depends on hydrophobicity, a mean pores dimension, and possibly temperature gradient across the GDLs. In this study, the GDL hydrophobicity is designed based on our previous study [2]. The GDL treated with 30 wt.% Polytetrafluoroethylene (PTFE) provide the best single AEMFC performance compared to the cases of 10, 20, or 40 wt.% PTFE treatment. In addition, for mean pore dimension (or average pore size), some research groups [3-7] have studied the correlation between average pore size of mean pore radius and water transport in GDLs through modeling or experiments. In fact, with the same materials and manufacturing process, the mean pore size will be significantly changed as different amount of weight percent of PTFE is treated. So, with the same PTFE treatment applied to three GDL samples in our work, their mean pore size could not be much different. In order words, we suppose that the mean pore size cannot be the main reason for the variation of cell performance. Besides, it is reported that temperature gradient across diffusion media induces additional water transport due to phase change [8-10]. In our work, the thermal conductivity of these GDL samples should be very similar due to the same materials and PTFE content treatment. Thus, the difference in temperature gradient across the GDLs can be caused by the different thicknesses of the GDLs. Although there has not been any report on the correlation between GDL thickness and gradient across the GDL, we also believe that there is a different temperature gradient across these GDL samples as installed in fuel cell stack, and thereby, influence on water transport. This is an issue should be addressed although the effect is maybe not significant because the different thickness is not large in terms of heat transfer (the largest difference is 180 mm for GDL-120 vs. GDL-310).

[1] Lin, G.; Nguyen, T. V. Effect of Thickness and Hydrophobic Polymer Content of the Gas Diffusion Layer on Electrode Flooding Level in a PEMFC. Journal of The Electrochemical Society 2005, 152, A1942.

[2] Truong, V. M.; Wang, C.-L.; Yang, M.; Yang, H. Effect of tunable hydrophobic level in the gas diffusion substrate and microporous layer on anion exchange membrane fuel cells. Journal of Power Sources 2018, 402, 301-310.

[3] Ahmad El-kharouf, Thomas J. Mason, Dan J.L. BrettBruno G. Pollet. Ex-situcharacterisation of gas diffusion layers for proton exchange membrane fuelcells. J. Power sources, 2012, 218, 393-404.

[4] Mehdi Mortazavi, Anthony D. Santamaria, Vedang Chauhan, Jingru Z. Benner, Mahbod Heidari, Ezequiel F. M edici. Effect of PEM fuel cell porous media compression on in-plane transport phenomena. Journal of Power Sources Advances, 2020, 1, 100001.

[5] M. Sepe,1, P. Satjaritanun, S. Hirano, I. V. Zenyuk, N. Tippayawong, and S. Shimpalee.  Investigating Liquid Water Transport in Different Pore Structure of Gas Diffusion Layers for PEMFC Using Lattice Boltzmann Method. Journal of The Electrochemical Society, 2020, 167 104516.

[6] Mehdi Shahraeeni, Mina Hoorfar. Pore-network modeling of liquid water flow in gas diffusion layers of proton exchange membrane fuel cells. International Journal of hydrogen energy, 2014, 1-13.

[7] Chang Sun Kong, Jooyong Kim, Hankyu Lee, Yong-Gun Shul, Tae-Hee Lee. Influence of pore-size distribution of diffusion layer on mass-transport problems of proton exchange membrane fuel cells. Journal of Power Sources, 2002, 108. 185-191.

[8] Yablecki J, Nabovati A, Bazylak A. Modeling the effective thermal conductivity of an anisotropic gas diffusion layer in a polymer electrolyte membrane fuel cell. J Electrochem Soc 2012, 159(6):B647–B653.

[9] Weber AZ, Hickner MA. Modeling and high-resolutionimaging studies of water-content profiles in a polymer-electrolytefuel-cell membrane-electrode assembly. Electrochim Acta, 2008,  53(26):7668–7674.
[10] Wang Y, Wang C-Y. A nonisothermal, two-phase model for polymer electrolyte fuel cells. J Electrochem Soc, 2006, 153(6):A1193–A1200.

Reviewer 2 Report

- You have to specify what "CL" means, in line 76, and not after (line 81);

- in lines 93 and 94 you assert that, to date, no study has been performed to correlate the thickness of the GDL and the AEMFC performance actually, there is a PhD thesis in which there an interesting study on this: “Water and Anion Transport in Electrochemical Devices” Travis John Omasta, Doctoral Dissertations, University of Connecticut Graduate School, 2018

- in line 147 you write “Two Teflon gaskets with appropriate thickness were also used…”, could the authors give some additional information about the thickness of the used gaskets?

- line 198 “…” what do you means? Probably, and other

- Could the authors explain better the evident decay of GDL-130 sample performance in fig. 4 in the resistive zone compared to GDL-230? The decay is too sudden, did the authors verify the status of the MEA post mortem electrochemical tests?

- Why was not a test performed by coupling a GDL-130 to the anode and GDL-310 to the cathode? This is because, as can be seen from fig. 5, when the thickness cathode increases, there is a slight improvement in the stability of power density curve

- In table 1, the authors insert values for the three GDLs of both air permeability and contact angle, but they are not commented on in the results and discussions chapter. How were the air permeability measurements performed? Do contact angle measurements make an important contribution to the paper? If yes, specify how, otherwise it is a parameter that can be deleted from the table

Author Response

Dear Reviewer,

We would like to thank the reviewer for their thoughtful review of the manuscript and provide valuable comments and suggestions. You raise important issues and your inputs are very helpful for improving the manuscript. We agree with your comments and we have revised our manuscript accordingly. All the changes are marked in red using the “Track Changes” function in Microsoft Word.

We hope that the reviewer will find our responses to their comments satisfactory, and we are willing to finish the revised version of the manuscript including further suggestions that the reviewer may have.

We thank the reviewer’s helpful comments and suggestions and provide the answers to specific comments below.

Comment 1:

- You have to specify what "CL" means, in line 76, and not after (line 81);

Answer to comment 1:

We thank the reviewer’s comment. This is our mistake. We have edited and the changes are marked in red in the manuscript, line 76.

Comment 2:

- in lines 93 and 94 you assert that, to date, no study has been performed to correlate the thickness of the GDL and the AEMFC performance actually, there is a PhD thesis in which there an interesting study on this: “Water and Anion Transport in Electrochemical Devices” Travis John Omasta, Doctoral Dissertations, University of Connecticut Graduate School, 2018.

Answer to comment 2:

We thank the reviewer’s comment. We missed Omasta’s PhD dissertation during preparing our manuscript although we have mentioned his work through his article (ref. 23 in the manuscript) in the Introduction section. From your comment, we have read carefully the thesis and we find out that although the balance and transport of water in anion exchange membrane fuel cell were investigated, the effects of gas diffusion layer (GDL) thickness on the cell performance was not reported in this dissertation. In particular, effects of some parameters is mentioned in the thesis as follows:

+ In chapter 2, Travis John Omasta examined the influence of the electrode and GDL as well as the flow rate and dew points of the anode and cathode gases on AEMFC performance. For the GDL, he employed Toray TGP-H-060 with different PTFE wetproofing (5 wt.% or 0 wt.% PTFE
wetproofing by weight).

+ In chapter 3, investigate the influence of electrode composition and structure, and operating conditions, on the transport properties of reactants, products, ions and water. In this study, only GDL Toray TGP-H-060 with 0 wt.% or 5 wt.% PTFE wetproofing was used again.

+ In chapter 4, the influence of anode catalyst layer thickness and catalyst distribution in lower loading electrodes is investigated. In this work, different catalyst layer thicknesses fabricated on the same GLD thickness (Toray TGP-H-060 with 5wt% PTFE wetproofing) were evaluated.

+ In chapter 5, highly active Pd-based cathode and anode catalysts were prepared and tested. The GDL Toray TGP-H-060 with 5 wt.% PTFE wetproofing was also employed in this investigation.

+ In chapter 6, he focused on the impact of CO2 and carbonate on the operation of anion exchange membrane (AEM) based electrochemical devices. The only type of the GDL Toray TGP-H-060 with 5 wt.% PTFE wetproofing was used again.

In in general, the influence of GDL thicknesses on AEMFC was still not investigated in the dissertation “Water and Anion Transport in Electrochemical Devices”.

Comment 3:

- in line 147 you write “Two Teflon gaskets with appropriate thickness were also used…”, could the authors give some additional information about the thickness of the used gaskets?

Answer to comment 3:

We thank the reviewer’s comment. The Teflon gasket thickness is used to provide a required 20-30% GDL compression. Specifically, the gasket thicknesses of 110 mm, 220 mm, and 250 mm were accompanied with the GDL-120 (120 mm of thickness), GDL-260 (260 mm of thickness), and GDL-310 (310 mm of thickness), respectively. With the measured catalyst layer thickness was around 20 mm, the amount of the compression of GDL-120, GDL-260, or GDL-310 is about 21.4%,  21.4%, or 24.2%, respectively.

All respective addition and changes are added to the section of the Single-cell AEMFC assembly and evaluation in the manuscript and marked in red, lines 176 -178.

Comment 4:

- line 198 “…” what do you means? Probably, and other

Answer to comment 4:

We thank the reviewer’s comment. We think that we are maybe misunderstanding this comment. However, based the sentence on line 198: “…causing a higher contact resistance between the catalyst layers and membrane surfaces.”, we would like to explain what we are talking about is that: From Figure 4, we observed that the power densities of GDL-120 are lower than those of GDL-260 at a current region from 400 – 1100 mA cm-2 and this region belongs to the ohmic loss. As presented in Table 1, we know that the through-plane resistance of GDL-120 (4.66 mW cm2) is smaller than that of GDL-260 (7.40 mW cm2) but the ohmic loss of GDL-120 is higher than that of GDL-260. So, it could be the higher contact resistance between the catalyst layers and membrane surfaces in the MEA made of GDL-120 because of the lower mechanical strength of GDL-120 compared to that of GDL-260.

Comment 5:

- Could the authors explain better the evident decay of GDL-130 sample performance in fig. 4 in the resistive zone compared to GDL-260? The decay is too sudden, did the authors verify the status of the MEA post mortem electrochemical tests?

Answer to comment 5:

We thank the reviewer’s comment. As seen in Figure 4, the power density of the MEA with GDL-130 suddenly drops at high current density only. This zone belongs to the gas transport limitation and it should be caused by the flooding in the anode or/and cathode electrodes. For each MEA, we repeated at least 5 times in different days and for each time, we operated over 50 cycles to examine the status of the MEA degradation in short-term operation. The obtained results were very similar for different times. Although we did not verify the status of the MEA by post-mortem analysis with electrochemical tests, we strongly believe that it is not the case of MEA degradation.

Comment 6:

- Why was not a test performed by coupling a GDL-130 to the anode and GDL-310 to the cathode? This is because, as can be seen from fig. 5, when the thickness cathode increases, there is a slight improvement in the stability of power density curve

Answer to comment 6:

We thank the reviewer’s comment. We agree with the reviewer that there is a slight improvement in the stability of power density curve at high current densities as comparing the obtained results of the MEA with GDL-130/GDL-130 and the MEA with GDL-130/GDL-260. However, there is still a flooding phenomenon observed at the similar high current level and it should be caused by the GDL-130 at the anode side. The reason why we did not perform a test for the case of MEA with GDL-130/GDL-310 is because we found that GDL-260 exhibited better cell performance than GDL-310 as inserted either anode or cathode electrodes (as seen in Figure 5(a)). Therefore, we believe that the cell performance of the MEA with GDL-130/GDL-310 cannot be better than that of the MEA with GDL-130/GDL-260.

Comment 7:

- In table 1, the authors insert values for the three GDLs of both air permeability and contact angle, but they are not commented on in the results and discussions chapter. How were the air permeability measurements performed? Do contact angle measurements make an important contribution to the paper? If yes, specify how, otherwise it is a parameter that can be deleted from the table

Answer to comment 7:

We thank the reviewer’s comment. The air permeability and contact angle are mainly discussed in the section 2.1 The preparation of gas diffusion layers.  In this study, we would like to investigate the effect of GDL thickness on the AEMFC performance. Thus, the GDL fabrication processes were kept the same for three types of GDL used in this work. These two measured results are used to examine some properties of these GDL samples after fabrication. In particular, the contact angle measurement is used to examine the hydrophobic property. The measured results show that the contact angle values of these samples are not much different, indicating that their hydrophobicity is similar at both back and microporous layer sides among samples. In addition, the air permeability measurement also confirms that the structure of the microporous layer (MPL) is similar because the air transport resistance of MPL is much higher than that of gas diffusion substrate.

Reviewer 3 Report

 Title: Effect of Gas Diffusion Layer Thickness on the Performance of 2 Anion Exchange Membrane Fuel Cells

 Authors: Van Men Truong, Ngoc Bich Duong, Hsiharng Yang

 Summary

The manuscript presents studies effects of GDL thickness on performance and water management of AEMFC. The authors investigated three GDLs with thicknesses of 120, 260, 310 microns. It is reported that application of 260 microns GDL for both electrodes ensured the best performance of AEMFC. Taking into account that AEMFC research is still in its infancy stage the work provides basic experimental results, which are important for building knowledge base for AEMFC operation. 

I noticed several minor misprints:

  1. Abstract, line 14: GDL26-0 should be changed to GDL-260.
  2. Results and discussion, page 5, line 163: GDL-230 should be replaced by GDL-260.
  3. Results and discussion, page 5, line 165: GDL-230 should be replaced by GDL-260.

Comments

  1. I was puzzled by the fact that the authors did not mentioned some publications discussing water management and effects of GDL on AEMFC performance. For example, J. Electrochem. Soc. 162 (2015) F483 (DOI: 1149/2.0131506jes). It was a special issue of J. Power Sources devoted to AEMFC back in 2018: https://www.sciencedirect.com/journal/journal-of-power-sources/vol/375/suppl/C. There are a lot of publications from many research groups discussing various aspects of AEMFC operation. I believe that some of these publications can be very useful for the data discussion and help improve the data analysis.
  2. 2. SEM images. I think that this figure could be a little bit bigger, otherwise it is difficult to distinguish details at cross-section images.
  3. MPL thickness should be indicated in the text. As I mentioned numbers are not readable at Fig. 2.
  4. I am wondering if the authors perform any break-in procedure for AEMFC?
  5. What was reproducibility of IV curves? The samples durability?
  6. Did you measure high-frequency resistance during IV recording?
  7. I would strongly suggest adding other electrochemical methods for evaluation of the AEMFCs like CV, EIS etc. These days presentation of IV curves is not enough for a paper to be published at modern journals.

Author Response

Dear Reviewer,

We would like to thank the reviewer for their thoughtful review of the manuscript and provide valuable comments and suggestions. You raise important issues and your inputs are very helpful for improving the manuscript. We agree with your comments and we have revised our manuscript accordingly. All the changes are marked in red using the “Track Changes” function in Microsoft Word.

We hope that the reviewer will find our responses to their comments satisfactory, and we are willing to finish the revised version of the manuscript including further suggestions that the reviewer may have.

 We thank the reviewer’s helpful comments and suggestions and provide the answers to specific comments below.

I noticed several minor misprints:

  1. Abstract, line 14: GDL26-0 should be changed to GDL-260.
  2. Results and discussion, page 5, line 163: GDL-230 should be replaced by GDL-260.
  3. Results and discussion, page 5, line 165: GDL-230 should be replaced by GDL-260.

We thank the reviewer’s notices. All the typos are corrected and marked in red in the manuscript, line 14, 203, 205, 207.

Comment 1:

I was puzzled by the fact that the authors did not mentioned some publications discussing water management and effects of GDL on AEMFC performance. For example, J. Electrochem. Soc. 162 (2015) F483 (DOI: 1149/2.0131506jes). It was a special issue of J. Power Sources devoted to AEMFC back in 2018: https://www.sciencedirect.com/journal/journal-of-power-sources/vol/375/suppl/C. There are a lot of publications from many research groups discussing various aspects of AEMFC operation. I believe that some of these publications can be very useful for the data discussion and help improve the data analysis.

Answer to comment 1:

We thank the reviewer’s comments. The reviewer is right in that we did not review some useful publications related to our study. We already read some of the suggested useful references. Although there are about 47 published articles in the special issue of J. Power Sources: Alkaline membrane fuel cells: state-of-the-art and remaining challenges, we found that some of them are really relevant to our manuscript. Thus, we have considered them as key references (ref. 23, 24, 26) in our manuscript and used some information to make our manuscript clearer.

All additions to the Introduction section in the manuscript are marked in red, line 83 – 94, 98 - 102.

[23] Kaspar, R. B.; Letterio, M. P.; Wittkopf, J. A.; Gong, K.; Gu, S.; Yan, Y. Manipulating Water in High-Performance Hydroxide Exchange Membrane Fuel Cells through Asymmetric Humidification and Wetproofing. Journal of The Electrochemical Society 2015, 162, F483-F488.

[24]  Luo, X.; Wright, A.; Weissbach, T.; Holdcroft, S. Water permeation through anion exchange membranes. Journal of Power Sources 2018, 375, 442-451.

[26] Reshetenko, T.; Odgaard, M.; Schlueter, D.; Serov, A. Analysis of alkaline exchange membrane fuel cells performance at different operating conditions using DC and AC methods. Journal of Power Sources 2018, 375, 185-190.

Comment 2:

SEM images. I think that this figure could be a little bit bigger, otherwise it is difficult to distinguish details at cross-section images.

Answer to comment 2:

We thank the reviewer’s comment. We agreed that it is not easy to distinguish details at cross-section images of these GDL samples at this magnification. In the manuscript, the cross-sectional SEM images are used to mainly show the structure inside the GDL samples and their thicknesses obtained from using measurement function in SEM operating software. With this purpose, we found that the magnification of the cross-sectional SEM images is the most suitable one because when the higher magnification of cross-sectional SEM images was used, it is hard to show the whole structure of the GDL. The main information that we can see from the cross-sectional SEM images is that the general structure of the GDL is a bilayer structure consisting of a layer of carbon fiber paper (gas diffusion substrate, GDS) and a layer of carbon particles (microporous layer, MPL). Basically, the GDS composes of carbon randomly distributed fibers and held together in a carburized resin matrix with PTFE coated on carbon fiber surfaces, whereas the MPL is a primary aggregate of carbon black particles with approximate diameters in the range of 20 – 50 nm. The porosity of the GDS is much larger than that of the MPL. Since the gas diffusion substrates have a heterogeneous structure, it is not easy to have the same or very similar cross-sectional SEM images of the different GDL samples.

Comment 3:

MPL thickness should be indicated in the text. As I mentioned numbers are not readable at Fig. 2.

Answer to comment 3:

We thank the reviewer’s comment. In this study, the MPL thickness is designed about 35 mm. The measured MPL thicknesses of GDL-120, GDL-260, and GDL-310 from cross-sectional SEM images are 36.1, 32.1, and 33.8 mm (as seen in Fig. 2(c), (f), (i)), respectively. The difference between the designed MPL thickness and the measured ones can be ascribed to the manufacturing tolerance of MPL thickness (about ± 2 mm) and the unsmooth GDS surface (as seen in Figure 2(a), (d), (g)).

All respective additions are added to the section of the preparation of gas diffusion layers in the manuscript and marked in red, lines 131 – 135.

Comment 4:

I am wondering if the authors perform any break-in procedure for AEMFC?

Answer to comment 4:

We thank the reviewer’s comment. For each test, a break-in procedure was first performed to obtain stabilization of the MEA before recording IV curves. The procedure is similar to Hamish A. Miller’s work [1]. In particular, the stabilization of the MEA was achieved by slowly ramping from 10 to 100 mA cm−2 (10 mA cm−2 increments, 1 min per step), followed by a 30 min hold at 100 mA cm−2; next, slowly again ramping from 100 to 400 mA cm−2 (10 mA cm−2 increments, 3 min per step), followed by a 30 min hold at 400 mA cm−2; and, then slowly ramping from 400 to 700 mA cm−2 (10 mA cm−2 increments, 3 min per step), followed by a 30 min hold at 700 mA cm−2. The polarization curves were taken from 1,1 to 0.2 volt in 0.02-volt increments for each cycle. The analyzed IV curves were collected from over 50 cycles at stable operating conditions.

All respective additions are added to the section of Single AEMFC assembly and evaluation in the manuscript and marked in red, lines 184 – 193.

[1] Hamish A. Miller, Francesco Vizza, Marcello Marelli, Anicet Zadick, Laetitia Dubau,
Marian Chatenet, Simon Geiger, Serhiy Cherevko, Huong Doan, Ryan K. Pavlicek,
Sanjeev Mukerjee, Dario R. Dekel. Highly active nanostructured palladium-ceria electrocatalysts for the hydrogen oxidation reaction in alkaline medium. Nano Energy 33 (2017) 293–305

Comment 5:

What was reproducibility of IV curves? The samples durability?

Answer to comment 5:

We thank the reviewer’s comment. The IV curves presented in the manuscript were the average results of each sample tested at least 5 times in different days and for each time, the fuel cell stack was operated over 50 cycles for recording the IV curves at stable conditions. The IV curves of each sample obtained from different times were very similar. In this work, all the results were just obtained at short-term tests. The sample durability will be evaluated in a long-term fuel cell test in next our work after we get more funding for this project.

Comment 6:

Did you measure high-frequency resistance during IV recording?

Answer to comment 6:

We thank the reviewer’s comment. We haven’t measured high-frequency resistance during IV recording. We knew that this method is actually a subset of the EIS method and in fact, we would like to do EIS analysis as suggested by the reviewer in order to provide a more detailed explanation of the variation of cell performance using different GLD samples. However, since we have not enough equipment, this EIS analysis will be our further work as we get more funding.

Comment 7:

I would strongly suggest adding other electrochemical methods for evaluation of the AEMFCs like CV, EIS etc. These days presentation of IV curves is not enough for a paper to be published at modern journals.

Answer to comment 7:

We thank the reviewer’s comment. We strongly agree with the reviewer that our work will be more qualified by adding other electrochemical methods for evaluation of the AEMFCs, especially, EIS method. EIS is an extension of the high-frequency resistance method that has been demonstrated to be a useful and powerful technique to study the different processes taking place in fuel cells. Based on the variation of the impedance with frequency, some important parameters including non-electrode ohmic resistance, electrode properties such as ohmic resistance and activation polarization resistance, double-layer capacitance, and transport properties can be studied. Due to the limitation of equipment and funding, we could not add EIS analysis data at the moment and this work will be done in our next project. To be honest, we hope that the reviewer is satisfied with the current results and accept to publish the manuscript in the special issue “Experimental Analysis and Numerical Simulation of Fuel Cells” of Processes.     

Round 2

Reviewer 1 Report

I accept the authors responses to my comments

Reviewer 2 Report

Dear Authors,

I would like to thank you for all the exhaustive corrections made. After reviewing the manuscript, I think it can be published in this form.

Best regards